# Comparative assessment of nutritional status in unoperated children with Congenital Heart Defects: Insights from a tertiary pediatric cardiac center in India

Radha Joshi[1]*, Manasi Bhoite[1], Poonam Mandhare[1], Shaoni Nath[1], Sudhir Kapoor[1], Rishikesh Wadke[2]*, Ragini Pandey[3]

**1** Sri Sathya Sai Sanjeevani Research Centre, Sri Sathya Sai Sanjeevani Research Foundation, Kharghar, Navi Mumbai, Maharashtra, India, **2** Department of Public Health, Sri Sathya Sai Sanjeevani Centre for Child Heart Care and Training in Paediatric Cardiac Skills, Navi Mumbai, Maharashtra, India, **3** Department of Paediatric Cardiac Surgery, Sri Sathya Sai Sanjeevani Center for Child Heart Care, Atal Nagar (Nava Raipur), Chhattisgarh, India,

* suv21688@gmail.com, drradha.joshi@ssssrf.org (RJ); drrishikesh.wadke@srisathyasaisanjeevani.org (RW)

## Abstract

Congenital Heart Defects (CHD) are structural cardiac malformations ranging from mild to severe forms; greatly impacting childhood mortality and morbidity. Malnutrition as comorbidity in CHD raises treatment complexity, lengthens post-operative recovery period and increases risk of developmental delays. This retrospective study evaluates patterns of malnutrition in 1678 unoperated CHD cases at out-patient department of tertiary pediatric cardiac centre in India compared to 11,894 community based controls. Z-Scores based on WHO reference charts were used for Weight for Age, Height for Age and Weight for Height calculations. Majority of CHD patients belonged to poor socioeconomic background [upper lower = 49.17% and lower middle = 42.99%]. 62.46% cases were underweight, 41.3% stunted and 53.93% wasted compared to controls showing 28.88% underweight (Z = 27.38, p < 0.01), 31.15% stunting (Z = 8.32, p < 0.01) and 14.04% wasting (Z = 39.01, p < 0.01), indicating highly significant undernutrition in cases compared to controls in same age group (0–6 years). Odds ratio analysis showed that CHD cases were 7.19 times more likely to undergo wasting, 4.19 times at higher risk of being underweight and were at 1.63 times increased risk of stunting than controls (p < 0.01). Pulmonary Arterial Hypertension (PAH) significantly exacerbated undernutrition in terms of wasting and underweight status in CHD (p < 0.01). Lower birth weight was found associated with undernutrition in CHD (p < 0.05). This first large-scale study from India comparing CHD patients with community controls provides comprehensive analysis of nutritional status in unoperated CHD cases indicating significantly higher undernutrition in CHD patients compared to non-CHD controls from same age group. This highlights the

**Data availability statement:** The datasets generated and/or analysed during the current study have been shared along with this paper as Supporting information.

**Funding:** The authors received no specific funding for this work.

**Competing interests:** The authors have declared that no competing interests exist.

need for comprehensive health screening in initial years of life which is crucial for early detection, timely CHD treatment and individualized nutritional management in pediatric cardiac care.

## Introduction

Congenital anomalies are fifth largest leading causes of death in early years of life [1]. Global report of March of Dimes (MOD) presents high prevalence of birth defects in India as 64.3 affected per 1000 live births [2]. Congenital Heart Defects (CHD) are amongst the most common anomalies present since birth and correspond to around 4-8 per 1000 live births [2] with high prevalence in Asia of 9 affected per 1000 live births [3]. CHD range from simple septal defects & shunt lesions like Atrial Septal Defect (ASD) to complex critical forms like single ventricle morphologies [4]. CHD etiology is complex which has prenatal and even periconceptional associations and there is interaction of several genetic or familial factors and environmental or non-genetic factors [5].

Children with CHD suffer significantly from malnutrition which grossly affects the increased morbidity and mortality reflected in poor surgical outcomes, prolonged post operative care, frequent hospital admissions, overall compromised growth & development and increased probability of death [6,7]. CHD children though may be born with normal anthropometric indices, soon postnatally suffer from high risk of undernutrition and failure to thrive. There are several causes of this undernutrition including the underlying defect itself, chromosomal abnormalities, insufficient calorie intake, hypermetabolism, feeding & swallowing problems and many more [8,9]. Malnutrition; mainly under nutrition is prevalent in children with CHD [10,11]. Childhood under nutrition is the primary cause of morbidities as it compromises immunity, increases risk of infections and prolongs suffering due to adverse prognosis [12]. Hence, in CHD patients malnutrition and ill health trigger a vicious cycle with gross impact on increased childhood mortality & morbidities.

Malnutrition is a result of inappropriate food intake, diet portions or improper absorption of nutrients. It can be due to either excessive loss or utilization of nutrients [12]. Undernutrition is a burning public health problem in India with almost 60 million underweight children [13]. As per WHO factsheets, malnutrition is the term that encompasses undernutrition (underweight, wasting, stunting), overweight, obesity, hidden hunger (micronutrient deficiencies) and consequent onset of noncommunicable diseases. It essentially means dietary imbalances or deficiencies due to several reasons. As per 2022 WHO data sheets, globally 45 million children were wasted (too thin for height), 149 million children <5 years old were stunted (very short for age) and 37 million were affected with overweight or obesity. Undernutrition can lead to around half of childhood deaths in children under 5 years. As per recent most comprehensive National Family Health Survey Data Report [14], 35.5% children in India under five are stunted, 19.3% are wasted, 32.1% are underweight with more children affected with undernutrition in rural population and 3.4% are overweight. Severe wasting is comparable in urban and rural children with 7.7% total children affected

with it in India. The predictors of childhood malnutrition have been enumerated in a recent report as birth weight, age, maternal education, infections, number of siblings, socioeconomic status of family and drinking water quality.

From literature we understand that India carries a significant dual burden of CHD and childhood undernutrition, yet their interrelationship remains poorly understood. Globally a few reports have documented poor nutrition status in CHD. But there are very few reports to compare observations in CHD cases with community based controls without CHD. The presented study is amongst the first reports from India to compare the nutritional status and undernutrition severity among CHD children from outpatient department (OPD) of a tertiary paediatric cardiac centre in Maharashtra, India with CHD-free controls from community. This study addresses a critical knowledge gap by providing novel evidence on the relationship between CHD and undernutrition, which may have direct implications for designing early screening strategies and tailored nutritional management in pediatric cardiac care.

## Materials and methods

### Study details

An observational retrospective case control study was conducted at Sri Sathya Sai Sanjeevani Centre for Child Heart Care & Training in Pediatric Cardiac Skills (SSSSCCHCTPCS), Kharghar, Navi Mumbai, Maharashtra which is a tertiary cardiac care centre providing totally free of cost complete healthcare to patients affected with CHD. We used all inclusive samples for cases and controls to eliminate selection bias. This retrospective study included analysis of n=1678 patient information from OPD of SSSSCCHCTPCS and records of n=11,894 non-CHD children in the same age group as controls from Public Health Department of the same Hospital, resulting in approximately 7 controls per case, ensuring sufficient statistical power for comparison of outcomes (Fig 1).

For cases, unoperated children with CHD were defined as those in the age group 0-6 years having isolated or complex structural cardiac abnormalities present since birth and who had not undergone any surgical procedure or catheter intervention for correction of the same. Anthropometric data of CHD patients visiting OPD of the hospital in the period June

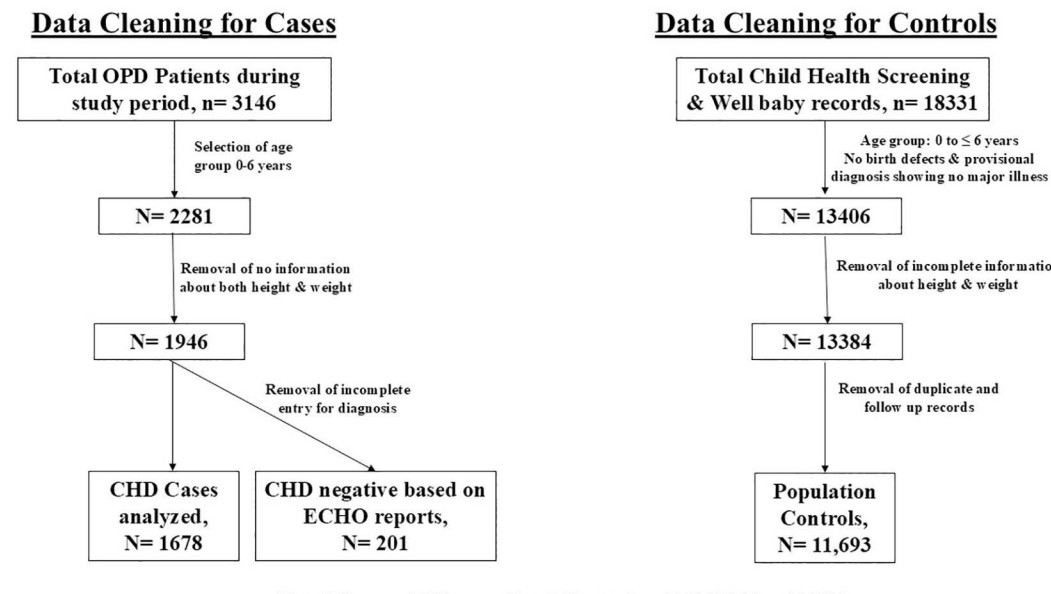

**Fig 1. Data cleaning for Nutritional Status analysis.**

2022- May 2023, was considered for this study. The study was approved by Institutional Ethics Committee- Sri Sathya Sai Sanjeevani Centre, Kharghar IEC (Registration No: EC/NEW/INST/2022/2782). Being a retrospective analysis, consent waiver was granted by IEC for this study.

Detailed multifactorial patient information collected in OPD of SSSSCCHCTPCS, as a part of CHD evaluation included reports of pre-operative physical examination and Echocardiography (ECHO) and self reported health history by the parents of the patients. Diagnosis of CHD was confirmed based on ECHO reports and such patients were classified as cases for this study. CHD were assigned International Statistical Classification of Diseases and Related Health Problems (ICD11) codes for ease of disease classification.

Controls were children in the same age group who were free from CHD and any major illness, chronic conditions, comorbidities, syndromes or any other congenital defects which might affect the child growth, as confirmed from detailed physical examination by experienced and qualified medical practitioners. Records of children screened at all anganwadis (community government health centres) of Panvel block, Raigad District, well baby clinics at Primary Health Centres (PHCs) and at SSSSCCHCTPCS in past five years till June 2023 were considered for this analysis. For anthropometric measurements of control children, the weight and height information were accessed from Grow Ayu (online database). Both case and control data sets were accessed for research from April 2024 after receiving all institutional and regulatory permissions. The authors did not have access to information that could identify individual participants during or after data collection. Incomplete entries or missing values in the data sets were not considered in respective analyses.

Weight was considered in kilograms and measured using weighing scale (sensitivity of 0.1 kilograms). Length or height in centimeters were measured using infantometer or stadiometer for toddlers and children.

## Statistical analysis

Data analysis and visualization was done using Microsoft Excel, Python and R software. Z-Scores based on WHO reference charts were used for Weight for Age (WAZ), Height for Age (HAZ) and Weight for Height (WHZ) calculations. Malnutrition was defined as WAZ, HAZ, and WHZ Z score ≤–2 SD or ≥2 SD. Z test for comparison of two proportions in cases and controls for WAZ, HAZ and WHZ was done. The significance was assessed using Chi-square test for categorical variables like underweight, stunting and wasting status. Odds ratio was calculated to check odds of undernutrition due to CHD. P-value < 0.05 was considered statistically significant. Categorical data were summarized as frequencies and percentages. To test for significant associations between categorical variables, we used the Chi-square test and Fisher's exact test, with a significance threshold set at p<0.05. Discrete variables like age and weight, were described using means and standard deviations. Differences in means were assessed using the Student's t-test. Data analysis was performed using Python and R software (version 4.2.2).

## Results

For cases, final analyzable dataset was arrived at based on the data cleaning done as shown in Fig 1. Total CHD patients were, n = 1678; whereas total controls (n = 11,894) included controls from community plus OPD patients with structurally normal heart. Since ECHO diagnosis was not available for suspected CHD cases from screening camps, we did not include them in case group.

The age and gender distribution of cases and controls is summarized in Table 1. In cases and controls, gender difference was significant, as confirmed from Chi- square test (($x^2$ = 6.20, p = 0.0127). Cases have 1.22 times higher proportion of males compared to females (OR= 1.22, p<0.05). Controls have a relatively more balanced male-female distribution. The case characteristics are described in Table 2. In the present study, most children with CHD belonged to Upper Lower class (49.17%) (Table 2).

Amongst all CHD cases, 62.46% were underweight, 41.3% stunted and 53.93% wasted compared to controls showing 28.88% underweight, 31.15% stunting and 14.04% wasting. There was significantly more wasting (Z= 39.01, p<0.01),

**Table 1. Age and Gender distribution of study participants.**

|  | Cases (n=1678) | Controls (n=11,894) |
|---|---|---|
| **Age Group (Years)** | 0-6 Years | 0-6 Years |
| **Age (Years) Mean±SD** | 2.95±1.70 | 2.77±1.65 |
| **Gender** |  |  |
| Males | 54.96% | 51.27% |
| Females | 45.04% | 48.73% |

**Table 2. Subject characteristics from OPD cases.**

| OPD Cases (n=1678) |  |
|---|---|
| **Location** |  |
| Maharashtra | 84% |
| Uttar Pradesh | 8.69% |
| Bihar | 2.47% |
| Other | 4.82% |
| **Acyanotic CHDs** | **74.55%** |
| Ventricular Septal Defect (VSD) | 34.39% |
| Atrial Septal Defect (ASD) | 20.68% |
| Coarctation of Aorta (CoA) | 1.49% |
| Patent Ductus Arteriosus (PDA) | 9.18% |
| Partial Atrio Ventricular Septal Defect | 0.360% |
| Miscellaneous | 8.46% |
| **Cyanotic CHDs** | **25.45%** |
| Tetrology of Fallot (TOF) | 12.63% |
| Transposition of Great Arteries (TGA) | 2.74% |
| Complete Atrio Ventricular Septal Defect | 1.97% |
| Total Anomalous Pulmonary Venous Connection (TAPVC) | 4.05% |
| Truncus Arteriosus | 0.30% |
| Single Ventricle | 1.78% |
| Tricuspid Atresia | 0.36% |
| Miscellaneous | 1.61% |
| **Intervention Type Advised** |  |
| Surgery | 33.31% |
| Cath | 8.05% |
| Medical Management | 58.34% |
| **Medical Management followed by surgery/ cath** | 0.3% |
| **Socioeconomic Class** |  |
| Upper | 0.62% |
| Upper Middle | 6.86% |
| Lower Middle | 42.99% |
| Upper Lower | 49.17% |
| Lower | 0.37% |

underweight children (Z= 27.38, p<0.01) and stunting (Z= 8.32, p<0.01) in case group as compared to controls from community in the same age group (0-6 years). The odds ratio indicated significantly higher risk of undernutrition among CHD cases, with 7.19-fold increased odds of wasting, 4.19-fold increased odds of being underweight and 1.63-fold increased

odds of stunting compared to controls. These findings indicate a substantially higher burden of undernutrition among children with CHD, particularly for wasting and underweight, underscoring the vulnerability of this group to poor nutritional outcomes. The mean height and weight across different age groups was fairly less in cases than controls (Figs 2 and 3).

As can be inferred from Table 3, mean height in cyanotic CHD patients was observed significantly lesser than acyanotic patients (p< 0.05) indicating pronounced stunting in this subgroup that may be attributed to the CHD severity. There was no significant difference observed in mean height/ length and weight between two genders (Table 3).

For WAZ, HAZ and WHZ analysis, minor fraction of data with missing values for either weight or height or out of range entries have been denoted as 'blanks'and have not been considered in respective analyses (Table 4). Measured underweight status, wasting and stunting was signficantly higher in CHD cases as compared to controls (p<0.01) using Z-test for comparing two proportions (Table 4).

Pulmonary Arterial Hypertension (PAH) is one the major complications in CHD and clinically is an important predictor for undernutrition. In CHD cases from OPD, 173 cases were reported to have mild to severe PAH; whereas the remaining 1505 cases did not show PAH. 82.66% of PAH cases were underweight, 61.83% showed stunting and 72.83% showed wasting (Table 5). Our data suggested signficantly higher odds of wasting (OR= 2.49, P<0.01) and being underweight (OR= 3.16, p<0.01) in CHD cases with PAH as compared to those who didn't develop PAH. Out of total 427 cyanotic CHD cases, 50 were reported to have PAH. Using Fisher's Exact Test, more undernutrition in terms of wasting (OR= 2.07; p=0.0325), and underweight status (OR= 3.17; p=0.0068), was observed in cyanotic CHD cases with PAH; though there was no signficant stunting in them (OR= 1.05; p>0.05).

Birth weight as one of the most important predictors of nutrition status was explored in our study to find an association of Low Birth Weight (LBW) with higher undernutrition in CHD cases. For controls data, LBW information was not available. In our study, 28.83% CHD cases reported LBW out of 1633 cases. Using Z-test for proportions, significantly higher

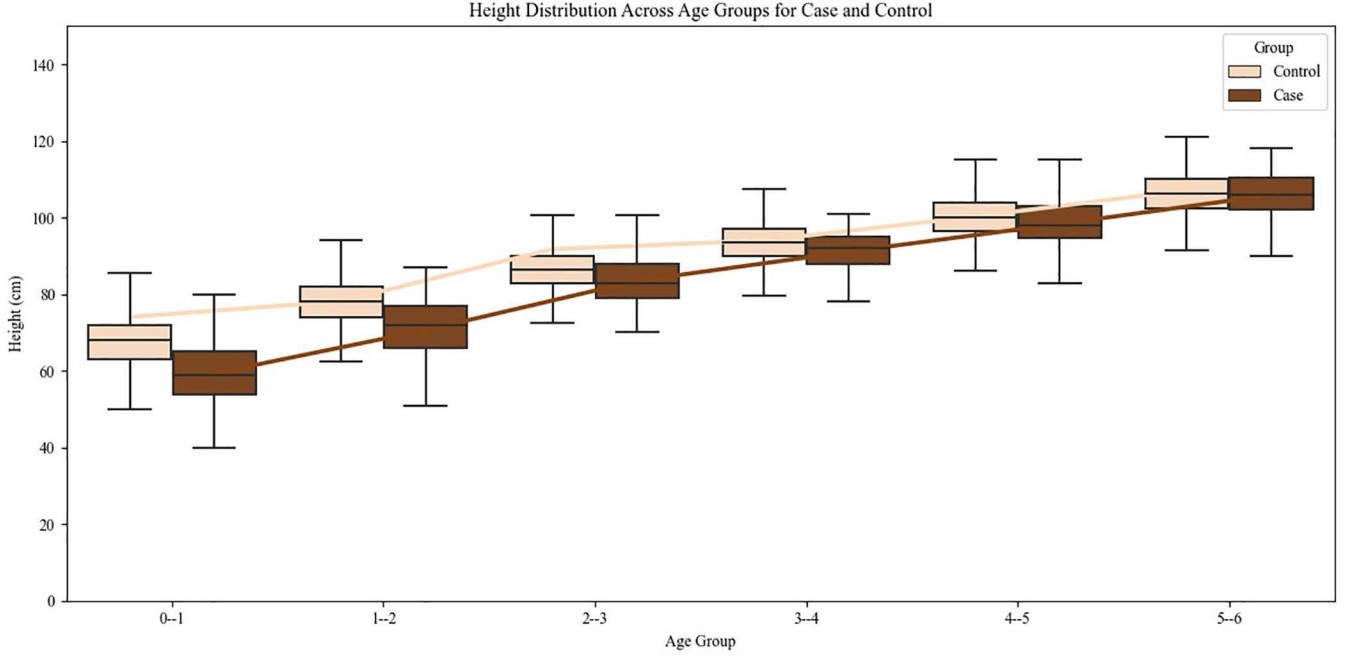

**Fig 2. Height Distribution across age groups in cases and controls.**

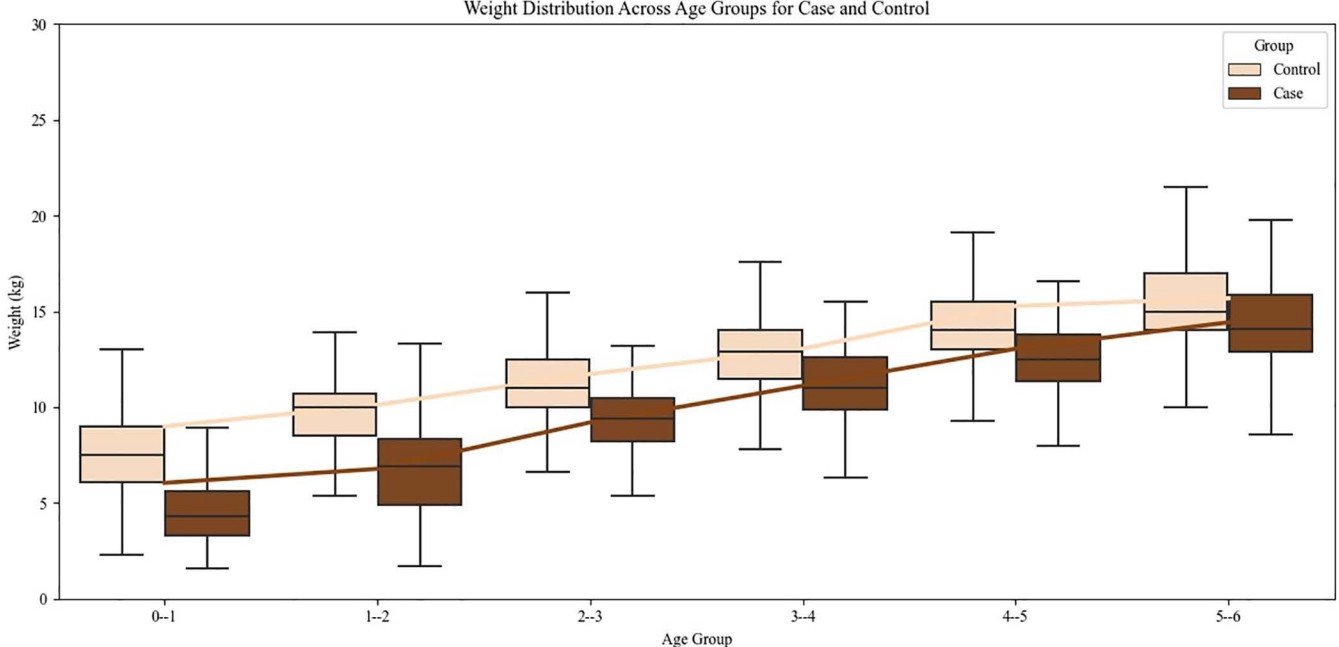

**Fig 3. Weight Distribution across age groups in cases and controls.**

**Table 3. Weight and Height variations based on Gender and CHD type.**

|  | Categories | Number of subjects | Mean | Standard Deviation | t- test value | P-value |
|---|---|---|---|---|---|---|
| **Weight (kg)** | Males | 919 | 7.88 | 13.92 | t = -0.48 | 0.63 |
|  | Females | 759 | 8.43 | 29 |  |  |
|  | Acyanotic CHD from OPD | 1251 | 8.61 | 25.4 | t = -1.63 | 1.10 |
|  | Cyanotic CHD from OPD | 427 | 6.6 | 3.49 |  |  |
| **Height (cm)** | Males | 919 | 71.26 | 18.03 | t = -0.18 | 0.56 |
|  | Females | 759 | 71.39 | 19.01 |  |  |
|  | Acyanotic CHD from OPD | 1251 | 72.27 | 18.92 | t = -3.87 | **0.0001** |
|  | Cyanotic CHD from OPD | 427 | 68.31 | 16.17 |  |  |

undernutrition in terms of wasting (Z = 4.66, P < 0.01), stunting (Z = 10.24, p < 0.01) and failure to thrive (Z = 8.65, P < 0.01) was observed in CHD cases with LBW (Fig 4).

## Discussion

Our research provided community-based prevalence of undernutrition in 0–6 years of age group. We observed 14.04% children thin for height (wasted), 31.15% short for age (stunted) and 28.89% children thin for age (underweight) in controls which is far less than reported population prevalence of undernutrition in children below 5 years of age in India (Wasting in 19% and Stunting in 36% children, 32% underweight children with 3% overweight as per NFHS-5 report) (p < 0.00001) [14] indicating better health status in general paediatric population in our study geographies.

A systematic analysis performed recently on a large dataset of 5,210,646 children of South East Asian origin from 345 studies & 25 survey datasets revealed that in year 2020, the indicators of undernutrition improved to 30.1% stunting,

**Table 4. Comparative Nutrition status in cases and controls.**

| Category | Wasting (%) | | | | | Stunting (%) | | | | | Underweight (%) | | | | | Total |
|---|---|---|---|---|---|---|---|---|---|---|---|---|---|---|---|---|
| | N | MAM | SAM | ON | BL* | N | MS | SS | Tall | BL* | N | MUW | SUW | OW | BL* | |
| Cases | 749 (44.64) | 366 (21.81) | 539 (32.12) | 19 (1.13) | 5 (1.13) | 860 (51.25) | 340 (20.26) | 353 (21.04) | 121(7.21) | 4 (0.24) | 592 (35.28) | 400 (23.84) | 648 (38.62) | 33 (1.97) | 5 (0.30) | 1678 |
| Controls | 9951(83.66) | 1178 (9.9) | 492 (4.14) | 246 (2.07) | 27 (0.23) | 7511 (63.15) | 2479 (20.84) | 1226 (10.31) | 678 (5.7) | – | 8139 (68.43) | 2372 (19.94) | 1063(8.94) | 320 (2.7) | – | 11894 |
| Analysis | Z=39.01; p<0.01 | | | | | Z=8.32; p=<0.01 | | | | | Z=27.38; p<0.01 | | | | | |

**Key:**

1. N: Normal,
2. MAM: Moderate Acute Malnutrition,
3. SAM: Severe Acute Malnutrition,
4. ON: Overnourished,
5. BL: Blanks,
6. MS: Moderate Stunting,
7. SS: Severe Stunting,
8. MUW: Moderate Underweight,
9. SUW: Severe Underweight,
10. OW: Overweight.

*Not considered in denominator for respective percentage calculations in each category of wasting, stunting and underweight.

**Table 5. PAH in CHD and nutrition status.**

| Category | | Underweight (%) | Stunting (%) | Wasting (%) |
|---|---|---|---|---|
| CHD | With PAH | 82.66 | 61.83 | 72.83 |
| | Without PAH | 60.99 | 40.93 | 51.76 |
| Acyanotic CHD | With PAH | 80.49 | 41.46 | 73.17 |
| | Without PAH | 56.91 | 37.68 | 50.53 |

### Comparison of Nutritional Status with Birth Weight

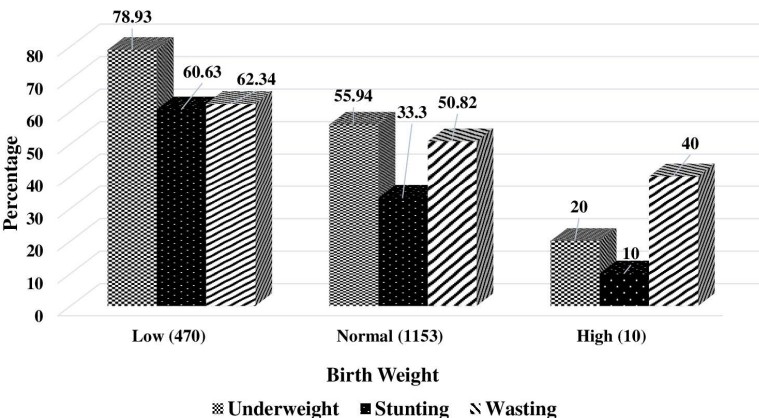

**Fig 4. Birth weight and nutrition status.**

10.8% wasting and 19.5% underweight in children from 5-19 years of age [15] which is still high as per WHO data sheets. According to WHO report 2020, 47 million children less than 5 years old are wasted with severe wasting in 14.3 million children,144 million children are stunted whereas 38.3 million children are overweight or obese [16]. Contrary to this, South Africa Demographic and Health Survey (SADHS) report of 2016 states wasting in 2.5%, underweight in 6% children under 5 years, but high stunting- 27% in them [16].

In CHD scenario, malnutrition prevalence ranges from 15-64% in CHD children [17]. Around 45% of total deaths in children below 5 years of age are attributed to malnutrition with 68.2% of deaths in early years of life in India. Cognitive and physical functions are found to be affected over the period due to malnutrition. 1% loss of height due to stunting in childhood is attributable to 2.4% loss of individual productivity [18]. This situation worsens with an underlying structural cardiac defect in terms of disease prognosis and post operative recovery. Malnutrition can be a determinant factor in etiology as well as adverse prognosis in CHD.

In our CHD cohort, majority of the patients belonged to upper lower and lower middle socioeconomic class as per Kuppuswamy scale which was used for classification in OPD since 2022 [19–21]. Under-representation of lower socioeconomic strata at our totally free of cost tertiary paediatric cardiac centre may be consequence of inadequately sought medical care due to financial constraints, lack of transportation, delayed diagnosis, lack of awareness or cultural barriers. This indicates yet prevailing healthcare access disparities, delayed referrals and systemic barriers in reaching this quality healthcare to masses.

Our study shows very high prevalence of undernutrition in the large CHD group scheduled to undergo corrective surgery or intervention, compared to community based controls in same age group. Our data presents 62.46% underweight cases affected with CHD in which 37.3% were severely underweight. These findings are similar to a previous article [17]

but is significantly higher as compared to recently published studies [10,11]. This prevalence matches with the expected prevalence of malnourishment in CHD cases from developed countries as per another previous study [22]. We observed wasting in 53.93% CHD cases, suggesting significant wasting than the stated global wasting prevalence of 7.5% in the age group less than 5 years [23]. This observed wasting in CHD cohort was nearly three times the wasting in children below 5 years in India (19.3%) as per NFHS-5 report. We observed no gender specific differences in height and weight in CHD patients (Table 3). Our data shows 41.3% stunted children affected with CHD whereas a lower stunting in 31.15% children in total controls; as well as higher wasting in 53.93% CHD cases as compared to controls (14.04%) (p<0.01).This data is alike the previously published reports presenting higher undernutrition in CHD cases as compared to age matched healthy controls [8,10,11,24]. Odds ratio analysis shows that CHD cases are 7.19 times more likely to undergo wasting, 4.19 times at higher risk of being underweight and have 1.63 times increased risk of stunting than controls (p<0.01). In this first large scale study from India, comparing institution-based patients with normal controls from the community, we thus present highly significant burden of undernutrition in CHD cases. CHD has associated failure to thrive condition, as proposed by several published studies [24]. As per standard definition, it refers to faltering weight and improper growth in pediatric patients where weight for age is two standard deviations less than mean (< 5th percentile). CHD patients may be born with normal birth weight but become undernourished as they grow, extent of which depends on the CHD type. Malnutrition in CHD patients is due to multiple factors including severity of CHD, cyanosis, multiple structural cardiac defects, PAH and more [10,23]. Inadequate nutrition triggers the vicious cycle of recurrent infections, consequent anemia and resultant undernutrition; and also leads to increased morbidities and even death in early years of life.

Causes of malnutrition in CHD patients are multifactorial [25]. Pathological CHD type was explored as one of these influencing factors [10]. We too observed significantly lower mean weight and stunting in cyanotic CHD patients as compared to acyanotic CHD cases similar to a recent study that reported higher wasting and stunting in cyanotic CHD cases [10]. Feeding inability is prominent is cyanotic CHDs with extended suck-rest cycles which can predispose them to undernutrition.

Another finding is that our study shows higher proportion of undernutrition in CHD cases than overnutrition. There was no significant difference between overweight children in cases (1.97%) and control (2.69%) groups as analyzed using Z test for two proportions (p=0.083).

Children with CHD who are undernourished are at increased risk of 30 days post operative mortality. Additionally in stunting cases, the use of inotropes and post operative hospital stay is observed to be higher [10]. Late diagnosis, unattended CHD due to various limiting factors like lack of expertise, availability of quality healthcare centres, affordability may lead to PAH or eventual heart failure [3,10].

CHD is commonly associated with complication of PAH. Incidence of CHD associated PAH is 2.2 and prevalence is 15.6 per million in CHD patients as previously reported in literature [26]. We found relatively high rate of PAH in CHD cases, i.e., 103 per 1000 CHD patients in our OPD data. This may be attributed to referral bias and case severity at our tertiary paediatric cardiac care centre. In our data, patient families reported lack of facilities and inaccessible quality health care for timely intervention for majority of patients before their arrival at our centre and several unreported underlying social, economic and cultural limiting factors in India in lines similar to as summarized in a concise study earlier [3].

Especially in CHD cases with unrepaired left to right shunt, persistent higher pulmonary pressures lead to vascular dysfunctions. This in turn leads to pulmonary vascular resistance and higher pressures in right side of heart finally causing right ventricular failure [27,28]. PAH causes dysponea and despite increase in energy needs, oral intake in patients with dyspnea gets considerably restricted due to improper coordination of breathing and swallowing. Apart from these, repeated hospitalizations in PAH patients may limit the age-required food intake ultimately reducing the calorie intake [29]. We observed wasting in 51.76%, stunting in 40.93% and underweight status in 60.99% of CHD cases without PAH; whereas in CHD cases with PAH group showed wasting in 72.83%, stunting in 61.83% and underweight in 82.66% cases (Table 5). The difference in nutrition status was prominent in both acyanotic and cyanotic CHDs with or without PAH,

except for no association of PAH was shown with stunting in cyanotic CHD cases (p < 0.05). Stunting reflects chronic undernutrition. In cyanotic CHD cases with PAH, life expectancy is markedly reduced, which may explain the observed lack of association between stunting and PAH, as these children might not survive long enough to develop stunting. One of the previous reports depicts PAH as one of the most important factors affecting nutrition status in CHD cases with findings similar to our study [25].

As per WHO criteria, LBW is defined as birth weight < 2.5 kg and it is one of the major predictors of prenatal mortality and morbidity. In Asia, more than 60% babies are born with LBW [30]. A few previous reports have presented that children with low birth weight are at greater risk of having undernutrition as they grow [30]. Our data further endorses this with prominent undernutrition in LBW CHD cases as compared to cases with normal birth weight (p < 0.05). The underlying CHD also predisposes the patients to poor intrauterine growth or spontaneous preterm birth as compared to normal babies and this is one of the primary causes of LBW. Coexistence of LBW and CHD in newborn further raises the risk of medical complications which further predisposes the patient to compromised nutrition status [31].

We would like to emphasize that all CHD are present since birth though diagnosed at different time points. Being retrospective study, data was available only for given time point. Thus, the nutritional trajectory for cases could not be assessed. Though this is a retrospective study design, we have used all-inclusive samples for both cases and controls thereby eliminating the limitation of selection bias. Also, inclusion of community-level controls enhanced the representativeness of study findings. While prospective studies offer certain advantages, the retrospective approach allowed us to analyze a large, real-world dataset that would have been challenging to obtain prospectively in the same timeframe. However we acknowledge that this study design restricted the availability of specific comparable variables which may not be routinely recorded, e.g., Specific exposures and predictors or effect modifiers like maternal health status, dietary habits, socioeconomic status or feeding practices, parental addictions and more; thereby limiting analysis based on potential confounders which might affect the nutritional status and CHD in offspring.

Our analysis revealed that males affected with CHD were 1.22 times more than female patients. Thus gender matching of cases and controls was not feasible as it would have led to substantial loss of valuable female control information. Although this investigation benefits from large comparative datasets, one more key limitation is that the case data is derived from a single institution, which may affect generalizability of findings. Another noteworthy point is that, although a significant association between CHD and undernutrition was observed, establishing causality remains a challenge. While periconceptional nutritional insufficiencies in mother may lead to abnormal cardiac development in utero, postnatally CHD may trigger undernutrition in the course of child growth and development due to feeding challenges and increased metabolic demands. Even with these limitations, this study holds merit in terms large sample size and its design as the first large-scale comparative analysis from India assessing the nutritional status of children with CHD in relation to controls from same age group. This robust dataset allows meaningful statistical comparisons and provides valuable evidence regarding nutrition in pediatric cardiac patients.

The insights from this study will be useful to coin future studies to delve deeper into growth and nutritional outcomes of children with CHD. To strengthen the evidence base, we recommend conducting prospective longitudinal studies to better assess causal relationships. Future research should also include detailed dietary assessments and socioeconomic details to account for additional influencing factors like duration of existing CHD. Furthermore, subgroup analyses based on CHD type and severity could help identify vulnerable populations and guide targeted interventions.

## Conclusion

This is a first of it's kind report from India to compare nutritional status in CHD with community based prevalence of undernutrition in children, in the age group of 0–6 years. This study highlights the effect of LBW and CHD complications like PAH on overall child growth and development. This report will be beneficial to bring much needed awareness for early detection and addressal of undernutrition in children, coupled with health assessment to detect presence of CHD,

consequent early referral for treatment and dietary management in order to improve disease prognosis and post operative recovery.

## Supporting information

**S1 Data. Supplementary material.**
(XLSX)

## Acknowledgments

We express our sincere gratitude to Chairman Dr. C. Sreenivas for all institutional support and guidance. We thank all Clinicians, data entry team, technical and non-technical staff of Sri Sathya Sai Sanjeevani Centre for Child Heart Care & Training in Pediatric Cardiac Skills, Kharghar, Navi Mumbai, Maharashtra for enabling data availability for research. We acknowledge our IT team for their technical assistance. We also thank the Public Health Department for their well-organized field work records.

## Author contributions

**Conceptualization:** Radha Joshi, Sudhir Kapoor, Rishikesh Wadke.

**Data curation:** Radha Joshi, Manasi Bhoite, Shaoni Nath.

**Formal analysis:** Radha Joshi, Manasi Bhoite, Rishikesh Wadke.

**Investigation:** Radha Joshi, Poonam Mandhare, Ragini Pandey.

**Methodology:** Radha Joshi, Rishikesh Wadke, Ragini Pandey.

**Project administration:** Radha Joshi.

**Resources:** Radha Joshi, Sudhir Kapoor, Rishikesh Wadke.

**Supervision:** Radha Joshi.

**Validation:** Radha Joshi, Rishikesh Wadke, Ragini Pandey.

**Visualization:** Radha Joshi, Manasi Bhoite, Shaoni Nath, Rishikesh Wadke.

**Writing – original draft:** Radha Joshi.

**Writing – review & editing:** Radha Joshi, Manasi Bhoite, Poonam Mandhare, Sudhir Kapoor, Rishikesh Wadke, Ragini Pandey.

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
