## [Decision Letter · Decision Letter 0]

24 Apr 2025

PGPH-D-25-00105

Population based Comparative Nutritional Status of Unoperated Congenital Heart Defects patients from a Tertiary Pediatric Cardiac Centre in India

Dear Dr. Joshi,

Thank you for submitting your manuscript to PLOS Global Public Health. After careful consideration, we feel that it has merit but does not fully meet PLOS Global Public Health’s publication criteria as it currently stands. Therefore, we invite you to submit a revised version of the manuscript that addresses the points raised during the review process.

We look forward to receiving your revised manuscript.

Kind regards,

Dickson Abanimi Amugsi, PhD

Academic Editor

Journal Requirements:

Additional Editor Comments (if provided):

Reviewers' comments:

Reviewer's Responses to Questions

**Comments to the Author**

1. Does this manuscript meet PLOS Global Public Health’s publication criteria?

Reviewer #1: Yes

Reviewer #2: Partly

2. Has the statistical analysis been performed appropriately and rigorously?

Reviewer #1: No

Reviewer #2: I don't know

3. Have the authors made all data underlying the findings in their manuscript fully available (please refer to the Data Availability Statement at the start of the manuscript PDF file)?

Reviewer #1: No

Reviewer #2: No

4. Is the manuscript presented in an intelligible fashion and written in standard English?

Reviewer #1: Yes

Reviewer #2: Yes

Reviewer #1: Comments on Manuscript PGPH-D-25-00105

Dear Respected Editor,

Thank you for the invitation to review this manuscript. The authors have undertaken important work addressing the nutritional status of children with Congenital Heart Defects (CHD) in India, a significant public health concern. The study provides valuable insights but requires substantial revision.

Overall

This study investigates the important issue of malnutrition in a large cohort of Indian children (0-6 years) with unoperated Congenital Heart Defects (CHD), comparing them to community controls. The findings highlight a significantly higher burden of underweight, stunting, and especially wasting in CHD cases. While the data are valuable, several major revisions are required.

Major Concerns

1. Study Design/Control Group

The claim of a "population-based" comparison is inaccurate. Controls are a community-based convenience sample (health camps/clinics) and may not represent the general population, potentially underestimating the true disparity (as suggested by comparison to NFHS data).

2. Missing Multivariate Analysis

The Methods state multivariate analysis was performed to identify independent factors associated with undernutrition, but these crucial results are missing from the manuscript.

3. Gender Matching Interpretation

The text claims gender matching (Line 151), citing a Chi-square p-value of 0.0127. This p-value indicates a statistically significant difference in gender distribution between cases and controls, contradicting the claim of matching.

Minor Concerns

1. PAH Interpretation

The comparison of high PAH prevalence in this tertiary center cohort (Line 270) with population incidence (Line 269) is potentially misleading. Clarify that the high rate reflects referral bias and case severity concentration.

2. Methodological Clarity

Briefly justify using RBSK charts for WHZ > 5 years instead of WHO 5-19y standards (Line 131). Ensure clear definitions distinguish the focus on undernutrition vs. broader malnutrition.

3. Data Presentation

Improve clarity of comparison labels (x, y, z, etc.) in Table 3. Ensure all abbreviations in Table 4 are clearly defined in the key.

Recommendation

Major Revision Required. The manuscript addresses a significant topic with valuable data, but requires substantial revision to address the concerns regarding study design description/limitations, inclusion of multivariate analysis, and correction of statistical interpretations before further consideration for publication.

Reviewer #2: The article meets publication criteria. The abstract succinctly summarizes the study’s objectives, methods, key findings, and implications. However, it could better highlight the novelty of the study (being the first large-scale Indian study comparing CHD patients with population controls).

The introduction effectively establishes the significance of CHD and malnutrition as public health issues, citing global and Indian prevalence data.

The justification for the study is strong, emphasizing the lack of Indian population-based comparisons and the clinical implications of malnutrition in CHD.

The study fills a gap by providing comparative data on malnutrition in CHD vs. healthy children, which is scarce in low-resource settings. The retrospective case-control design is appropriate for comparing nutritional status between CHD patients and controls. Data from a tertiary cardiac center, ensuring clinical validity. Population-based controls from health camps improve generalizability.

Statistical Analysis, Appropriate use of Chi-square, t-tests, and multivariate analysis. However, details on adjustments for confounders (e.g., socioeconomic status) could be clearer.

Limitations- The retrospective design may introduce selection bias, Lack of detailed dietary intake or feeding difficulty data. Controls from health camps may not fully represent the general population.

This study provides valuable insights into malnutrition in Indian CHD patients, with robust methodology and significant public health implications. However, its retrospective nature and lack of granular data on dietary and socioeconomic influences limit deeper mechanistic insights. The findings underscore the need for integrated nutritional interventions in CHD care.

Recommendations

- Prospective longitudinal studies to assess causality.

- Inclusion of dietary and socioeconomic factors.

- Subgroup analysis by CHD type and severity.

**Do you want your identity to be public for this peer review?** For information about this choice, including consent withdrawal, please see our Privacy Policy

Reviewer #1: **Yes: ** Nimo Mohamoud Barakaale

Reviewer #2: No

---

## [Decision Letter · Decision Letter 1]

2 Sep 2025

PGPH-D-25-00105R1

Comparative Assessment of Nutritional Status in Unoperated Patients with Congenital Heart Defects: Insights from a Tertiary Pediatric Cardiac Center in India

Dear Dr. Joshi,

Thank you for submitting your manuscript to PLOS Global Public Health. After careful consideration, we feel that it has merit but does not fully meet PLOS Global Public Health’s publication criteria as it currently stands. Therefore, we invite you to submit a revised version of the manuscript that addresses the points raised during the review process.

EDITOR: Please insert comments here and delete this placeholder text when finished. Be sure to:

Please ensure that your decision is justified on PLOS Global Public Health’s publication criteria  and not, for example, on novelty or perceived impact.

We look forward to receiving your revised manuscript.

Kind regards,

Dickson Abanimi Amugsi, PhD

Academic Editor

Journal Requirements:

Additional Editor Comments (if provided):

Reviewer #1: The reviewer has recommended that the manuscript be accepted for publication. However, they noted several minor issues that need your attention before moving forward. Addressing these concerns will improve the quality of the manuscript and ensure it meets publication standards.

Reviewer #3: The reviewer highlighted several important issues that could significantly impact the reliability of the study's findings. These concerns may influence the interpretation of the results as well as the overall conclusions drawn from the research. I recommend that you take their feedback seriously and address it thoughtfully.

Reviewer #4:The reviewer has recommended that the manuscript be accepted for publication. However, they noted some minor issues that need your attention before moving forward. Addressing these concerns will improve the quality of the manuscript and ensure it meets publication standards.

Reviewers' comments:

Reviewer's Responses to Questions

**Comments to the Author**

Reviewer #1: (No Response)

Reviewer #3: (No Response)

Reviewer #4: All comments have been addressed

publication criteria?

Reviewer #1: Yes

Reviewer #3: No

Reviewer #4: Yes

3. Has the statistical analysis been performed appropriately and rigorously?

Reviewer #1: Yes

Reviewer #3: No

Reviewer #4: Yes

4. Have the authors made all data underlying the findings in their manuscript fully available (please refer to the Data Availability Statement at the start of the manuscript PDF file)?

Reviewer #1: Yes

Reviewer #3: Yes

Reviewer #4: Yes

5. Is the manuscript presented in an intelligible fashion and written in standard English?

Reviewer #1: Yes

Reviewer #3: Yes

Reviewer #4: Yes

Reviewer #1: Comments on Manuscript PGPH-D-25-00105RI

Clarity of Gender Matching:

There is a discrepancy with regard to gender matching.

According to Line 39 of the updated abstract, "age (0-6 years) & gender matched controls."

However, it is evident from the data (Table 1, Lines 150–152) that "Gender difference was significant in cases and controls... Cases have 1.22 times higher proportion of males... (p=0.0127)."

Lines 311-313 of the discussion further suggest that "gender matching of cases and controls was not feasible."

Statistical Reporting and Data Presentation:

The statement "indicating significant undernutrition in cases (p<0.05) compared to age (0-6 years) matched controls" is followed by the chi-square value "(χ2= 6.68, p=0.009)" which is bit unclear. To what particular comparison is this chi-square applied? Is it a test for one specific indicate or for variations in nutritional status overall? The significance for WAZ, HAZ, and WHZ derived by the Z-tests mentioned later for Table 4 may be clearer to understand.

Table 4: "Blanks" Give a brief explanation of how "Blanks"—missing data for WAZ, HAZ, or WHZ—were handled in the percentage calculation in the table footnote or methods. For the specific Z-score calculation, were they not included in the denominator?

Minor Suggestions:

Correct typographical error: "Congenial" to "Congenital" (Line 30).

Line 244 (Discussion): The Kuppuswamy scale citations (21)(22)[(23) – the last (23) is outside the bracket. Correct formatting.

Recommendation:

While the study addresses a relevant public health issue and offers potentially useful conclusions, there are important issues that need to be addressed which I mentioned comments.

Reviewer #3: The authors raised an important and interesting topic—studying undernutrition in children with congenital heart defects (CHDs) using an observational retrospective case-control design (Cases: Children diagnosed with CHDs. Controls: Children without CHDs).

Major issues:

• The issue of reverse causality is significant in such kind of study; it is difficult to determine whether CHDs led to undernutrition or vice versa.

• The rationale for the investigation is not well justified in the background section.

• The eligibility criteria for cases (children with CHDs), and the methods of case ascertainment and control selection, were not adequately described. There is no explanation of who the cases are or how the authors assessed the children with CHDs (methods section). Furthermore, how controls were assessed is not described—which is very important. Controls should not have chronic conditions or syndromes that affect growth.

• The matching criteria were unclear; specifically, the reason for gender matching was not justified, and the number of controls per case was not described.

• The authors did not clearly mention the exposures/predictors, potential confounders, or effect modifiers in their study, even though they used retrospective data.

• Despite the retrospective nature of the study, unadjusted estimates (such as odds ratios) and, if applicable, confounder-adjusted estimates (AORs) were not reported. It would be helpful to specify which confounders were adjusted for and why they were included in the final analysis. There is a possibility of confounding, and the authors did not clearly specify how they controlled for confounding variables. Important demographic and clinical variables—such as socio-economic status, feeding practices, and comorbid infections—that may affect nutritional status should have been adjusted.

• There is inconsistent use of the terms “undernutrition” vs. “malnutrition.”

• The findings may not be generalizable beyond the specific population or setting, especially as the study is not representative (one institution ..Sri Sathya Sai Sanjeevani Centre for Child Heart Care & Training in Pediatric Cardiac Skills)

Reviewer #4: This study addresses a significant and under-researched topic, particularly in a resource-limited setting where delayed surgical intervention is common. The focus on "unoperated" patients is a major strength, as it captures a population uniquely vulnerable to the long-term metabolic and hemodynamic consequences of CHD. The topic is of great relevance to pediatric cardiologists, nutritionists, and public health professionals working in similar contexts.

However:

1. The term "unoperated" needs precise definition. Were these patients awaiting surgery? Were they deemed inoperable? Was this their first presentation to the center? The timing of the assessment relative to the diagnosis is crucial. A child diagnosed years ago will have a different nutritional trajectory than one diagnosed recently. This potential confounding factor must be mentioned and discussed.

2.Please define the abbreviations (CHD, VSD, ASD, etc.) in a footnote of Table 2 for readers outside the field.

**Do you want your identity to be public for this peer review?** For information about this choice, including consent withdrawal, please see our Privacy Policy

Reviewer #1: **Yes: ** Nimo Mohamoud Barakale

Reviewer #3: No

Reviewer #4: **Yes: ** Abdul Nazer Ali

---

## [Editor Report · Decision Letter 2]

14 Sep 2025

Comparative Assessment of Nutritional Status in Unoperated Children with Congenital Heart Defects: Insights from a Tertiary Pediatric Cardiac Center in India

PGPH-D-25-00105R2

Dear Jadha,

We are pleased to inform you that your manuscript 'Comparative Assessment of Nutritional Status in Unoperated Children with Congenital Heart Defects: Insights from a Tertiary Pediatric Cardiac Center in India' has been provisionally accepted for publication in PLOS Global Public Health.

Best regards,

Dickson Abanimi Amugsi, PhD

Academic Editor